# Peer review of "Monitoring of Peroxide in Gamma Irradiated EVA Multilayer Film Using Methionine Probe"

_polymers, 2020, doi:10.3390/polym12123024_

Round 1
Reviewer 1 Report
In addition to the scientific, this manuscript has a practical significance as well, because it develops methods for defining the storage conditions in single-use bags (dry and with the contents). The experiments are well designed, the results are clearly presented and the following few things should be changed:
- Lines 152 and 153. "Figure 3. Plot of storage time vs concentration of methionine oxide generated to exemplify the three responses used in the experimental design." The figure shows the methionine sulfoxide. The names given at the figure and figure signature must to be harmonized.
- In the Table 2 is given "Oxidized methionine concentration". The unit for the concentration of oxidized methionine is not shown here. This need to be supplemented.
- Lines 125 and 126. "Scheme 2. Formation of methionine sulfoxide." The scheme does not show the formula of methionine and its oxidation. Scheme with the exact formula of methionine need to be written.
Author Response
In addition to the scientific, this manuscript has a practical significance as well, because it develops methods for defining the storage conditions in single-use bags (dry and with the contents). The experiments are well designed, the results are clearly presented, and the following few things should be changed:
- Lines 152 and 153. "Figure 3. Plot of storage time vs concentration of methionine oxide generated to exemplify the three responses used in the experimental design." The figure shows the methionine sulfoxide. The names given at the figure and figure signature must be harmonized.
We changed the name: Figure 3. Plot of storage time vs methionine sulfoxide concentration generated to exemplify the three responses used in the experimental design.
- In the Table 2 is given "Oxidized methionine concentration". The unit for the concentration of oxidized methionine is not shown here. This need to be supplemented.
We added the unit: µM.
- Lines 125 and 126. "Scheme 2. Formation of methionine sulfoxide." The scheme does not show the formula of methionine and its oxidation. Scheme with the exact formula of methionine need to be written.
We modified the scheme of the methionine in the article.

Reviewer 2 Report
This is a very nice work. The experiments are well planned and the results are very well presented.
I suggest that the authors present a comparison between methionine and homocysteine.
I want the authors to check whether the same result, i.e., sulfoxide formation is observed in homocysteine.
This will illustrate the role of the methyl group bound to sulfur in these results.
I want to review the revised manuscript.
Author Response
In article[1], Du et al. showed that the cysteine is readily oxidized in several compounds. Detection and titration of cysteine oxidation products is not straightforward; therefore, homocysteine is expected to react in the same way as cysteine.
For us, the role of methyl in methionine is to control the oxidation process (i.e. formation of methionine sulfoxide and methionine sulfone readily detected by HPLC), in sharp contrast to oxidation of cysteine (see Figure 1).
Thus, as literature describes the reaction of cysteine with peracetic acid (same reactivity is expected for homocysteine as for any linear alkyl thiols) and as the benefit of such experiment is not obvious for the article, the experiments suggested are not performed. Nevertheless, if the Editor think that such experiments are crucial for the article, we will comply.
Figure 1: Proposed reaction pathways for cysteine oxidation by peracetic acid.
[1] P. Du, W. Liu, H. Cao, H. Zhao, C.-H. Huang, Oxidation of amino acids by peracetic acid: Reaction kinetics, pathways and theoretical calculations, Water Research X. 1 (2018) 100002. https://doi.org/10.1016/j.wroa.2018.09.002.

Round 2
Reviewer 2 Report
Accept in the present form